# Flexible Implementation of the Trilinearity Constraint in Multivariate Curve Resolution Alternating Least Squares (MCR-ALS) of Chromatographic and Other Type of Data

**DOI:** 10.3390/molecules27072338

**Published:** 2022-04-05

**Authors:** Xin Zhang, Romà Tauler

**Affiliations:** 1Department of Chemistry, Capital Normal University, Beijing 100048, China; xinzhang@cnu.edu.cn; 2Institute of Environmental Assessment and Water Research (IDAEA-CSIC), 08043 Barcelona, Spain

**Keywords:** chemometrics, chromatography, three-way data, MCR-ALS, trilinearity

## Abstract

Multivariate Curve Resolution Alternating Least Squares (MCR-ALS) can analyze three-way data under the assumption of a trilinear model using the trilinearity constraint. However, the rigid application of this constraint can produce unrealistic solutions in practice due to the inadequacy of the analyzed data to the characteristics and requirements of the trilinear model. Different methods for the relaxation of the trilinear model data requirements have been proposed, like in the PARAFAC2 and in the direct non-trilinear decomposition (DNTD) methods. In this work, the trilinearity constraint of MCR-ALS is adapted to different data scenarios where the profiles of all or some of the components of the system are shifted (not equally synchronized) or even change their shape among different slices in one of their data modes. This adaptation is especially useful in gas and liquid chromatography (GC and LC) and in Flow Injection Analysis (FIA) with multivariate spectroscopic detection. In a first data example, a synthetic LC-DAD dataset is built to investigate the possibilities of the proposed method to handle systematic changes (shifts) in the retention times of the elution profiles and the results are compared with those obtained using alternative methods like ATLD, PARAFAC, PARAFAC2 and DNTD. In a second data example, multiple wine samples were simultaneously analyzed by GC-MS where elution profiles presented large deviations (shifts) in their peak retention times, although they still preserve the same peak shape. Different modelling scenarios are tested and the results are also compared. Finally, in the third example, sample mixtures of acid compounds were analyzed by FIA under a pH gradient and monitored by UV spectroscopy and also examined by different chemometric methods using a different number of components. In this case, however, the departure of the trilinear model comes from the acid base speciation of the system depending on the pH more than from the shifting of the FIA diffusion profiles.

## 1. Introduction

The Multivariate Curve Resolution-Alternating Least Squares (MCR-ALS) method was designed approximately 30 years ago [1] for the factor decomposition (matrix factorization) of two-way data using a bilinear model and of three-way with the optional application of the trilinearity constraint. The implementation of this trilinearity constraint in the ALS algorithm is very flexible and it allows for the handling of cases where the profiles of some of the components in one of the data modes are shifted. This situation often happens in datasets obtained in the simultaneous analysis of multiple samples by hyphenated chromatographic spectroscopic analytical methods. In this work, some of the advantages of this flexible implementation of the trilinearity constraint are shown and the results obtained with other multiway data analysis methods are compared.

Most multicomponent systems measured using spectroscopic multivariate responses and sensors can produce multiple data matrices which can be analyzed by chemometric methods based on a bilinear model factor decomposition (matrix factorization) of the measured data. These multicomponent systems include chemical reactions [2], industrial processes [3], chromatography [4], spectroscopic images [5], food [6], biological [7] agricultural [8], medical [9], and environmental samples [10]. This bilinear model factor decomposition or matrix factorization has been widely used for the analysis of different types of data due to the great variety of analytical instruments [1] producing bilinear data such as the multivariate extension of the linear Beer’s law in UV-Vis absorption spectroscopy.

From its initial development, MCR-ALS has been extended to the simultaneous analysis and resolution of multiset data coming from samples analyzed in different experiments or at different experimental conditions [1,11,12]. Multiset data can be simultaneously analyzed with the extension of a bilinear model applied to row- or/and column-wise concatenated (augmented) data matrices. By using proper constraints like non-negativity, unimodality, closure, local rank constraints and others, MCR-ALS performs the bilinear factor decomposition of the multiset data. Depending on the common information shared among the simultaneously analyzed multiset data, column-wise or/and row-wise matrix types of augmentation can be used to improve the results of MCR-ALS. When the profiles of the same component in the extended mode of the multiset data are synchronized and have the same shape, a trilinear type of constraints can be applied on MCR-ALS to conform with the extension of the bilinear model to the corresponding trilinear type of models. With the development of hyphenated analytical instruments, multiway datasets can be acquired very fast and the corresponding multilinear type models can be applied accordingly. The trilinear models built for multiway data have the advantage of eliminating the rotation ambiguities associated to bilinear type of modes and therefore provide unique resolution solutions [13,14,15]. Recent works about the rotation ambiguity problem associated with quantitative estimations by MCR methods can be found in the literature [16,17]. The profiles obtained for each component using a trilinear model are unique and provide an efficient and simple way to explain the observed variance in the raw data [14].

Moreover, results obtained by MCR-ALS with trilinearity constraints have been shown to be equivalent to those obtained by other trilinear model resolution methods like the alternating trilinear decomposition (ATLD) [18], parallel factor analysis (PARAFAC), [19] and other extension methods [20,21,22]. Other multilinear decomposition methods have been proposed in the literature for the extension of the trilinear model to the quadrilinear and even higher multilinear models in the analysis of data from multi-hyphenated instruments [14]. However, in the application of all these methods for resolution purposes, the data should fulfill the underlying multilinear model for all the components obtained in the factor decomposition. When the data are not conforming the trilinear (or multilinear) model, the use of these methods produces the wrong resolution results. In some cases, a preprocessing oriented approach is proposed to correct for the lack of fulfillment of the trilinear model, for instance in the case of peak misalignments of the profiles of the same component in the different simultaneously analyzed datasets, 2D warping methodologies or similar methods have been proposed [23,24]. In some other situations, like in liquid chromatography, peak shape changes can occur when strong coelutions are present [25,26,27,28,29]. In these situations, the strict trilinear model is not valid and the data resolution with these methods produces unique profiles far from the true profiles. Due to these frequent experimental data variability sources, PARAFAC2 [30,31], direct non-trilinear decomposition (DNTD) [32], ATLD-MCR [33], and MCR with the soft-trilinearity constraints (a modification of the trilinearity constraint allowing for smaller deviations of the perfect trilinear behavior in the profiles constrained) have been proposed [34]. Similarly, in MCR-ALS, the trilinearity (or its multilinear extension) constraint can be implemented separately for the different components in their different modes, and it can also be implemented at different levels in mixed multilinear models. Until now, these types of mixed multilinear models have not been proposed in other multiway data analysis methods [25]. A general scheme has been proposed for selecting the appropriate data processing model according to the properties exhibited by the multi-way data set under study [35].

In this work, a simulated liquid chromatography-diode array detection (LC-DAD) dataset was first analyzed to show the use of MCR-ALS with the variant of the trilinearity constraint where the profiles of the different components can be shifted among the different datasets, i.e., they do not need to be synchronized. A second data set example was obtained by the gas chromatography-mass spectrometry (GC-MS) analysis of wine samples [36], where apart from peak shifting, there was also a significant contribution of a background signal changing from run to run. Finally, a third example consists in the flow injection analysis (FIA) of a mixture of analytes having different acid-base properties and speciation. In all these cases, the results obtained by the different methods were compared. In particular, the flexible application of the trilinear constraint and of mixed bilinear trilinear models with MCR-ALS in different data scenarios is shown in detail. The MCR-ALS results were compared with those obtained by ATLD, PARAFAC, PARAFAC2, DNTD and with those obtained in previous studies in the literature [25,32].

## 2. Methods

### 2.1. MCR-ALS Bilinear Method

Multivariate Curve Resolution Alternating Least Squares, MCR-ALS, is a chemometrics method frequently used to resolve the concentration and spectra profiles of the chemical constituents present in mixtures of unknown composition using only the information contained in the experimental data measured by different types of analytical instruments [11,12,37], arranged in a data matrix.

MCR-ALS, performs the bilinear factorization of the experimental data according to Equation (1) (in element-wise form) and (2) (in matrix form):(1)$$x_{\mathrm{ij}}=\overset{N}{\underset{n=1}{\sum }}c_{\mathrm{in}}s_{\mathrm{jn}}+e_{\mathrm{ij}\;}\;$$
**X** = **C**
**S^T^** + **E**(2)

This bilinear factor decomposition of the data matrix **X** (I, J), with I rows (samples) and J columns (variables, wavelengths or channels) is achieved by an iterative alternating least-squares optimization of the two factor matrices **C** (I, N) and **S^T^** (N, J), where N, the number of factors or components, is much lower than I and J. In MCR mixture analysis chemical problems, the **C** factor matrix is commonly associated to the concentrations of the sample constituents or chemical species and the **S^T^** factor matrix is commonly associated to the spectra/signals of these chemical constituents.

The ALS optimization in MCR-ALS minimizes the Frobenius norm of **E** (residuals/errors) using an alternating least squares algorithm under constraints [13,38,39,40]. The first step in this optimization is to estimate the number of components contributing to the data variance in **X**. This can be initially estimated from the number of singular values associated to the systematic data variance of **X**, excluding the experimental error [41,42,43]. With this estimation of the number of components, an initial estimation of either **C** or **S^T^** factor matrices is searched. This can be performed in different ways, for instance selecting the most different row or columns profiles of the data matrix (like in the purest variables obtained using the SIMPLe-to-use Interactive Self-modeling Mixture Analysis, SIMPLISMA method [44,45]. ALS optimization iterations are then performed under a suitable set of constraints on **C** and **S^T^** profiles. The most common constraints applied in MCR bilinear model decompositions are the non-negativity constraints of **C** and **S^T^** factor matrices, for instance using non-negative least squares approaches [46]. Other constraints can be implemented during the ALS optimization like closure, unimodal, selectivity and local rank [40], which were not considered in this work.

MCR-ALS bilinear modelling can be easily extended to the simultaneous factorization of multiple data sets or matrices. This multiset data can be arranged in different ways according to the data mode (way or direction), which is kept in common in their simultaneous analysis. In the case of this work, the three data examples investigated were arranged in a column-wise data matrix augmentation and the extension of the bilinear model of Equation (2) can now be written as:(3)$$X_{\mathrm{aug}}=\left[\begin{array}{c} X_{1}\\ X_{2}\\ X_{3}\\ \vdots \\ X_{K} \end{array}\right]=\left[\begin{array}{c} C_{1}\\ C_{2}\\ C_{3}\\ \vdots \\ C_{K} \end{array}\right]S^{T}+\left[\begin{array}{c} E_{1}\\ E_{2}\\ E_{3}\\ \vdots \\ E_{K} \end{array}\right]=C_{\mathrm{aug}}S^{T}+E_{\mathrm{aug}}$$
where **X**_aug_ is the column-wise augmented data matrix composed by the **K** data matrices **X_k_** (I_k_, J). from the different data sets (samples, slices) simultaneously analyzed, vertically concatenated. In the case all data matrices have the same number of rows I (as in the data examples investigated in this work), the dimensions of **X**_aug_ are (I × K, J). In this data arrangement, the column vector space of **X_aug_** is shared for all the individual data matrices **X_k_**.

**C**_aug_ (I × K, N), is the augmented factor matrix having the K individual factor (concentration) matrices also vertically concatenated, **C_k_** (I, N), for the N components, and **S**^T^(N, J), is the common factor matrix with the spectra of these N components. The ALS factorization of **D_aug_** is performed in the same way as for the individual data matrices, but with the possibility of adding additional constraints related to the multiset data structure and related to the common components (chemical speciation) in the different individual data matrices [1,13,14,25,37,40,47,48,49,50]

### 2.2. MCR Trilinear and Mixed Bilinear-Trilinear Methods

The bilinear model factor decomposition or factorization of a two-way data matrix (Equation (3)) can be further extended to the trilinear model factor decomposition or factorization of the data in multiple individual data matrices arranged like in a three-way data set, **X **(I, J, K), with three data modes (ways or directions) representing the measured data along them. In this work, these three data modes are the concentration (with a retention time axis), the spectral (with a wavelength/mz axis), and the sample (with a sample index axis. The trilinear model factor decomposition can be written element-wisely (compare with Equation (1)) as:(4)$$x_{\mathrm{ijk}}=\overset{N}{\underset{n=1}{\sum }}c_{\mathrm{in}}s_{\mathrm{jn}}z_{\mathrm{kn}}+e_{\mathrm{ijk}\;}\;$$

For a trilinear model with N components, the data element x_ijk_ in **X** can be represented as the sum of the product of three elements, c_in_, s_jn_ and z_kn_, (i = 1, …, I, j = 1, …, J, and k = 1, …, K), where n is the considered component (n = 1, …, N). c_in_, s_jn_ and z_kn_ are the elements of the concentration, **C** (I, N), spectra, **S** (J, N), and sample, **Z** (K, N) factor matrices in their ith, jth and kth rows and nth column, respectively.

The trilinear model can be also written in matrix form according to the following Equation (compare with Equations (2) and (3)) as:(5)$$X_{k}=C\;Z_{k}\;S^{T}+E_{k}$$

According to this Equation, every data matrix **X**_k_ (also called slices) of the three-way data set **X** is decomposed with the same concentration, **C** (I, N) and spectral, **S**^T^, factor matrices, and they differ in their relative amounts depending on the data slice k considered, which are stored in the elements of the diagonal matrix **Z_k_** (N, N) (i.e., $C\;Z_{k}$.can be considered also as **C_k_).** The diagonal elements of these **Z_k_** matrices can be grouped in the third factor matrix of the trilinear model decomposition, **Z** (K, N).

In MCR-ALS, three-way data sets can also be decomposed under the assumption of a trilinear model and the profiles in the three data modes, in **C**, **S^T^** and **Z**, can be obtained. However, this trilinear model data decomposition is achieved in a different way than by the other trilinear decomposition methods such as PARAFAC or DTLD (see below). In the case of MCR-ALS, an additional trilinearity constraint can be applied during the MCR-ALS bilinear factor decomposition of the augment data matrix **X**_aug_. This implementation of the trilinearity constraint in MCR-ALS is performed algorithmically and it is shown graphically in Figure 1 and as pseudocode in the Appendix B. According to this implementation, the trilinear modeling of the profiles of the augmented **C**_aug_ matrix in MCR-ALS is applied in a flexible way, separately and optionally for the profiles of every component during the ALS optimization. Therefore the main advantage of this implementation is that it can be applied component-wisely, which allows performing the decomposition considering different model complexities, like all the components being trilinear, some components being trilinear and others bilinear, or all of them being bilinear. The application of the trilinearity constraint to all the components for three-way data, as shown in Figure 1, implies that the concentration (elution) profiles of all the components in the different **C_k_** matrices, have the same shape and that they are totally synchronized (blue arrows in Figure 1).

Figure 1 According to this trilinearity constraint, every augmented concentration profile in **C**_aug_ (I × K, N)_,_ for the component n, **c_aug, n_** (I × K, 1), is folded as a one component matrix **C****_n_** (I × K) with I rows (number of rows in each data slice **X_k_** and **C_k_**) and K columns (number of samples simultaneously analyzed) by setting **c_n1_,** …, **c_nk_** individual concentration profiles of this component in the different slices one besides the other. This **C_n_** (I, K) one component concentration profiles matrix is then analyzed by singular value decomposition (SVD). Only the first component is considered in the decomposition giving the common concentration profile **c_n_** (I, 1) and the K samples profile **z**^T^ (1, K), Both vectors are then multiplied using their Kronecker product [51], which recalculates the long one component concentration profile $\hat{c}$**_aug,n_** (I × K, 1), which fulfilled the required trilinear model requirements of equal shape and synchronization of the $\hat{c}$**_n1_****,** …, $\hat{c}$**_nk_** individual concentration profiles. When this is performed for all the components, n = 1, …, N, the **C_aug_** (I × K, N) is updated and the ALS iteration proceeds. The clear advantage of this implementation of the trilinearity model is that it can be applied component by component, it is flexible and fast. This also implies that mixed bilinear-trilinear models are feasible, i.e., the trilinear condition is applied for some of the components but not for other. This implementation of the trilinear model has already been generalized to quadrilinear, to pentalinear or in general to multilinear models as shown in [14,52]. When ALS finally converges and the final MCR solutions are obtained, **C** is made up of the concentration (column) profiles of the different components (N), which are the same for all data K matrices simultaneously analyzed, whereas **Z^T^** is made up of the profiles giving the relative amounts of these components in the different K matrices (sample profiles). Therefore, from the application of MCR-ALS trilinear, the trilinear three factor matrices in the three data modes, **C**, **S^T^** and **Z^T^** are obtained from the decomposition of **X_aug_**. These factor solutions obtained by this implementation of the trilinear model in MCR-ALS are analogous to the ones obtained by other trilinear based models like PARAFAC, as it was already shown in previous works [25,27,53].

In addition, the application of the trilinearity constraint during the ALS optimization also has the possibility to consider departures of the trilinearity model due to the lack of synchronization among the peaks of the concentration profiles of the same component in the different individual data matrices (slices), although they are still preserving the feature of equal shape among them. This problem has been also considered in the literature by other methods like in PARAFAC2, or by using a data pretreatment method which tries to realign the raw data profiles (not the resolved component profiles) using special techniques like in icoshift [54] or correlation optimized warping (COW) [55]. However, in case of strong overlapping among the component profiles, these approaches can fail [26]. The MCR-ALS trilinear algorithm is easily adapted to correct for the lack of synchronization of the concentration profiles in **C** factor matrix at each ALS iteration, as it is shown in the lower part of Figure 1 (orange arrows) and in the Appendix B. The trilinear model is preserved because the shapes of the concentration profiles of the same component in the different data matrices are kept to be the same, but they are shifted appropriately during the ALS optimization based on the optimal bilinear decomposition of the raw data. When this variant of the trilinearity constraint is applied, the shifts in the peak maxima of the profiles of the same component in the different matrices, **c_aug_**(I × K, 1), are first corrected (synchronized), before the application of SVD. Once the augmented profile, **c’_aug_** (I × K, 1), is rebuild from the Kronecker product of the first singular vectors, **z’_n_** ⊗ **c’,** the position of the peak maxima are restored to their original position and the new augmented profile $\hat{c}$**_aug_** (I × K, 1) is updated again for the next ALS iteration. As for the regular trilinearity constraint with synchronization, in this case the application of the trilinearity constraint without synchronization (orange arrows) can be applied for all or for only some of the components. This implementation of the trilinearity constraint is extremely flexible and it allows to build mixed bilinear-trilinear models with and without synchronization, which will cover many of the situations encountered in practice, in real world three-way data analysis situations. The presence of shifts in concentration profiles happens frequently in chromatography due to the slight variations and lack of reproducibility of the experimental conditions among different chromatographic runs. In the present work, changes in peak retention times (peak shifts) are investigated in different examples for LC-DAD, GC-MS and FIA data. The pseudocode for the implementation of the trilinearity constraint code is given in the Appendix B for the two cases examined here with (Section B) and without (Section A) shift correction.

As already explained, the main advantage of the trilinearity constraint with/without synchronization implementation in MCR-ALS is that this constraint can be applied independently and optionally for the different components resolved during the general MCR-ALS bilinear decomposition allowing building mixed bilinear-trilinear models. This gives much flexibility to data analysis, allowing for the implementation in MCR-ALS of full bilinear and trilinear models, mixed bilinear-trilinear models with and/or without profile synchronization (peak shifting). This flexible application of the trilinearity constraint for every component individually, and the possibility of correcting for unsynchronized (shifted) profiles in one of the data modes during ALS, allowing for different combination of models in the same data analysis, is investigated in this work. The different possibilities are coded in the following way, for every component of the model, 0 means that its profile is modelled in a bilinear way, 1 in a trilinear way, and 2 in a trilinear profile with shift correction. For instance, in the case of a 3 component model, (0,0,0) indicates that all three components are modelled in a bilinear way, (1,1,1) in a trilinear way, and (2,2,2) in a trilinear way with shift correction. However, more flexibility is allowed when the different components of the same system are modelled in a mixed way. For instance, in the model (2,2,1) the two first components are modelled in a trilinear way with shift correction and the third component modelled in a trilinear way without shift correction. And in the case, for instance, of (1,1,0), the two first components are modelled in a trilinear way, but the third one is modelled in a bilinear way. Profiles modelled in a bilinear way can differ among different data slices both in synchronization and shape. This allows the flexible implementation of bilinear and trilinear models in MCR-ALS, which could be extended also to more complex multilinear and other multiway interaction models [14].

### 2.3. Other Trilinear Methods (ATLD, PARAFAC, PARAFAC2 and DNTD)

Results obtained by the different variants of MCR-ALS are compared in this work with those obtained by other trilinear model-based methods when they are applied to the same data system, such as ATLD [18], PARAFAC [19], PARAFAC2 [30] and DNTD [32]. PARAFAC performs the trilinear model factorization of the three-way data set applying an alternating least squares optimization of the factor matrices in the three modes. by successively assuming the factor matrices in two modes known and then estimating the unknown factor matrix in the third mode [56,57]. In fact, the ALS factorization is performed in a similar way as in MCR-ALS, but in the later case this is only performed for one of the three modes and therefore only requires the fulfillment of the bilinear model. ATLD is like PARAFAC but instead of using ALS it uses a generalized singular value decomposition method to calculate the required pseudoinverses. ATLD is claimed to converge faster than PARAFAC and to be less sensitive to the number of resolved components [18]. In both cases, PARAFAC and ATLD, the analyzed data should fulfill the premises of the trilinear model to get the correct matrix factorization. Every component is defined by a triad of unique vector profiles. Different from PARAFAC and ATLD, PARAFAC2 allows the profiles of the components in one of the two factor modes to differ somewhat among the data slices [18,36]. In PARAFAC2, each data slice of the three-way data sample, **X_k_** (I, J) is modeled according to Equation (6):(6)$$X_{k}=C_{k}\;Z_{k}S^{T}+E_{k},\;\;k\;=1,\;\ldots ,\;K\;$$
where **X**_k_ (I, J) represents the data slice (data matrix) k. and **C_k_** (I, N), **Z_k_** (N, N) and **S^T^** (N, J), have the same meaning as those in the trilinear model of Equation (5), except that now the factor matrix **C_k_** may be different for the different data slices **X_k_**. In contrast, **S^T^** is still the same for all of the data slices. However, in PARAFAC2 **C_k_** are not freely modelled like in a bilinear model, but they are constrained to have their cross product **C^T^_k_C_k_** to be constant over all the data slices. The application of this constraint keeps the uniqueness of the decomposition and gives the chance to obtain vector profiles in the different **C_k_** matrices differing in their synchronization, for instance in the case of chromatography allowing the elution profiles of the same component being time shifted in the different data slices (chromatographic runs).

DNTD is similar to PARAFAC2, although the algorithm is not ALS and the application of the cross-product constraint is performed differently. The shifting profiles in **C_k_** are regularized by their average profile in all data slices which controls the convergence and accuracy of the shifting profile. If the regularization parameter is set to 1, the method is almost equivalent to the PARAFAC. On the contrary, if it is set to 0, no regularization is imposed, and the results are closer to those of the MCR-ALS bilinear model. As iterations progressed, the regularization varies gradually from 1 to 0 driven by the improvement in the data fitting. More details of this algorithm can be found in [23]

### 2.4. Quality Parameters

To evaluate the quality of the data fitting finally achieved after application of the different methods applied in this work, the following parameters have been considered.

The first is the lack of fit (lof), which is defined as:(7)$$\mathrm{lof}\;(\%)=100\;\times \;\sqrt{\frac{\sum_{\mathrm{ijk}}d_{\mathrm{ijk}}-\hat{d_{\mathrm{ijk}}}}{\sum_{\mathrm{ijk}}d_{\mathrm{ijk}}^{2}}}$$
where d_ijk_ is one of the data elements of the three-way data **X**, and ${\hat{d}}_{\mathrm{ijk}}$ is the corresponding recalculated element of this data matrix by ALS. lof gives a measure of the data fit quality in relative terms, in the same units as the measured data, and comparable with estimations of the experimental relative error.

Another parameter used is the explained data variances, R^2^ which is calculated as:(8)$$R^{2}=100\;\times \;\left(1-\frac{\sum_{\mathrm{ijk}}e_{\mathrm{ijk}}^{2}}{\sum_{\mathrm{ijk}}d_{\mathrm{ijk}}^{2}}\right)$$
where e_ijk_ are the elements of the **E** matrix and d_ijk_ are the elements of the raw data set **X**.

One easy way to test if a model is more adequate than others for the description of a particular data set is to compare the values of lof (%) and R^2^ obtained when these different models are applied. For instance, for a particular three-way data set when the bilinear and trilinear models are compared, two situations are usually encountered in practice. When the trilinear model is adequate to the data structure, lof (%) and R^2^ fitting values obtained applying the bilinear or the trilinear model will be similar, apart from the larger overfitting ubiquitous tendency of bilinear models. In contrast, if the trilinear model is not adequate to model the structure of the investigated data set, the lof (%) and R^2^ fitting values obtained applying the bilinear and trilinear models will differ considerably, in an amount which cannot be associated to the overfitting tendency of bilinear models. Although intermediate situations can occur in practice due to experimental noise and conditions, this comparison of lof (%) and R^2^ fitting values obtained applying different type of models can be very helpful in practice in the final selection of the more appropriate model to analyze the investigated data. This comparison should be used complementary to the examination of the chemical significance of the resolved profiles obtained by the application of the different type of models. When the trilinear model is not adequate to explain the investigated data set, some of the resolved profiles (if not all) will have unreasonable shapes from a chemical point of view, and this observation can be used to decide the adequacy of the postulated model. In addition, if some reference profiles are available from the literature, from libraries or from previous chemical knowledge, as it may occur with spectral (i.e., 2nd mode) or sample (i.e., 3rd mode) profiles, they can be compared with those obtained by the application of the different type of models calculating the correlation coefficient (r^2^) between them. This will give a measure of their similarity, and from them the vector angles between the two profiles compared can be obtained as defined by the two following Equations:(9)$$r^{2}=\frac{xy^{T}}{\left\|x\|\|y\right\|}$$
(10)$$\mathrm{Angle}=\frac{180}{\pi }\times \;\arccos\left(\frac{xy^{T}}{\left\|x\|\|y\right\|}\right)$$
where **x** and **y** are the two vector profiles to compare. The calculation of r^2^ (Equation (9)) and angle (Equation (10)) for the case of the augmented unsynchronized (shifted) profiles is performed directly, without considering separately the two embedded data modes derived from the trilinear model. This type of comparison is especially relevant for the case where the profiles in one of the modes are not synchronized like in chromatography when peaks are time shifted, or in a bilinear model when different modes are enclosed in a single profile, i.e in the row mode of column-wise augmented data matrices, like in Equation (3) for **X_aug_** and **C_aug_** (see in the Section 4)

### 2.5. Testing the Adequacy of the Trilinear Model by the Singular Value Decomposition of the Data Matrices Augmented in Their Different Modes

Three-way datasets **X **(I, J, K) can be reshaped in three possible two-way augmented data matrices according to their three different data modes (ways or directions). The individual two-way data matrices can be concatenated vertically in the column-wise augmented data matrix, **X_caug_** (I × K, J), can be concatenated horizontally in the row-wise augmented data matrix **X_raug_** (I, J × K) and can be vectorized and concatenated in the slice-wise data matrix **X_saug_** (K, I × J). When the trilinear model is adequate for the analysis of the three-way data set **X **(I, J, K) and there is no degeneracy, rank overlap nor rank deficiency in any of the three modes [13,14,25] (the application of Singular Value Decomposition, SVD, [43]) to **X_caug_**, **X_raug_** and **X_saug will_** produce in the three cases a similar number of significant larger singular values compared to those associated to experimental noise. In other words, the chemical rank (mathematical rank in absence of noise, [25]) should be equal to the number of chemical sources of data variance in the three cases. This is an easy test that helps with the investigating of the three-way data structure in many circumstances. Different causes produce departures of this ideal situation. In chemistry, and especially for spectroscopic data, every chemical species is defined by a single spectrum profile, and different chemical species have different spectra, and their mixture will be the linear sum of them weighted by their relative concentrations. These species spectra are in many circumstances unique and do not change due to experimental conditions. Therefore, **X_caug_** usually preserves the chemical rank of the system and can be used to estimate the number of chemical species present in the analyzed data, assuming that each one of them has a unique spectrum and that their relative concentrations in the samples change independently (see [25] for discussion of different possible scenarios). However, this very desired situation is not so frequently encountered in the other two types of data matrix augmentations, **X_raug_** and **X_saug_**. In these cases, the chemical rank can be either lower (rank deficiency) or higher (rank augmentation). One example of the first case is for **X_saug_** when the number of simultaneously analyzed slices (samples) is small and/or there is not total independency among them, for instance because the changes in the total concentrations of the constituents in the different samples are linearly dependent (rank degeneracy). One example of the other case, when the chemical rank is augmented and higher than the number of chemical species present in the analyzed system, is when the profiles of these chemical species change in one of the modes of the different data slices (samples) analyzed. This situation frequently occurs, for instance in chromatography, where the time (elution) profiles of the components are not unique and they change for the different data slices (samples) analyzed, either because they are not time synchronized and consequently they are shifted, or even worse because their shapes also change (for instance due to coelution and column overloading [25,26,28] In these cases, especially in the latter, the data does not conform with the trilinear model and the SVD of the **X_raug_** matrix will reflect this situation, with a chemical rank higher than expected and higher than the chemical rank of **X_caug_**. This situation will be investigated in the data sets of this work, particularly for data examples 1 and 2 in relation to the lack of synchronization of the chromatographic elution profiles.

In summary, different scenarios can be considered. (1) In the case of a data set fulfilling the standard trilinear model, the SVD of **X_caug_**, **X_raug_** and **X_saug_** will give the same number of significant singular values (larger than those associated to experimental noise), which will be equal to the number of chemical species contributing to the data variance in the analyzed sample; (2) In the case of a data set showing a rank deficiency or/and degeneracy in one of the data modes (usually the third slices/samples mode), the SVD of **X_caug_**, **X_raug_** will give the right number of significant singular values, equal to the number of chemical species present in the sample, but the SVD of **X_saug_** will give a number of significant singular values lower than the number of chemical species present in the sample. (c) In the case of a data set not fulfilling the trilinear model, the SVD of the different augmented data matrices will differ. For instance, when the SVD of **X_raug_** is larger than the SVD of **X_caug_**, and larger than the number of chemical species present in the sample, it happens when there is a lack of synchronization in the profiles of the components in one of the modes (for instance in the time/elution profiles as in the data examples below). When the problem is only in the time axis shifting of the profiles, the system can still be considered a soft departure of the standard trilinear model, and different methods have been proposed to circumvent the problem maintaining the trilinear model properties (see, for instance, DLTD, PARAFAC2 and variants of MCR-ALS with the shifted adapted trilinearity constraint).

### 2.6. Software and Calculations

In this paper, all tests and calculations were performed using an Intel Core i7-6500U 2.50 GHz computer running 64-bit Windows 10. All calculations have been performed under the MATLAB computation and visualization environment version R2010 (The MathWorks, MA, USA). PARAFAC and PARAFAC2 [30] calculations have been performed using their versions downloaded from the internet at http://www.models.kvl.dk/algorithms (Accessed on 1 October 2021) [48]. The ATLD algorithm was obtained from the MVC2 toolbox, which can be freely downloaded from [58,59]. DNTD was downloaded from https://github.com/JinZhangLab/DNTD [32]. The MCR-ALS toolbox was downloaded from http://mcrals.wordpress.com/download/mcr-als-toolbox [40]. Some of the mixed models including trilinear unsynchronized profiles were implemented in this work in a command line version of MCR-ALS, which can be obtained under request to RT.

## 3. Data

### 3.1. Synthetic LC-DAD Data

This data set is used to test and compare the results obtained by different data analysis methods proposed in the literature under controlled conditions. The three-way data present significant deviations of the trilinear model due to the lack of synchronization (large peak maxima shifts) of the profiles in one of the data modes. This synthetic data set is mimicking multiset data obtained by LC-DAD analysis of a set of samples, giving elution profiles with strong peak shifts, i.e., they are not synchronized. The three-way dataset **X** has 600 rows (retention times), 96 columns (wavelengths), and 11 data slices, one for each analyzed sample (or third mode elements). A total number of eleven samples were analyzed by LC-DAD. Elution profiles of the three sample compounds are plotted in Figure 2A, showing the presence of large shifts in their chromatographic peaks. The UV spectra of these three components are plotted in Figure 2B. The relative compositions of these eleven samples are plotted in Figure 2C.

### 3.2. Wine GC-MS Experimental Data

A GC-MS wine dataset consisting of 44 samples of red wines produced from a single wine grape variety was downloaded from http://www.models.kvl.dk/Wine_GCMS_FTIR.(Accessed on 1 October 2021) The aroma profiles of the samples were measured using a dynamic headspace GC-MS technique. The details of the dataset and measurements can be found in the literature [36]. Significant shifts of the retention time peaks and the presence of a fluctuating background signal were observed. The dataset analyzed in this work is focused only on the chromatographic elution time region between 16.52 min and 16.76 min, where 3-hydroxy-2-butanone and hexyl acetate were eluting [36]. The resulting three-way dataset **X** has 25 rows (retention times), 200 columns (mass units) and 44 data slices, one for each analyzed sample. The experimental mass spectra of 3-hydroxy-2-butanone and hexyl acetate are known and can be downloaded from MassBank [60] (http://www.massbank.jp/Index, accessed on 1 October 2021).

### 3.3. Flow Injection Analysis (FIA) Experimental Data

The third dataset analyzed in this work is a data set obtained in the flow injection analysis (FIA) of a chemical mixture of different acid-base isomers investigated in a previous work [61]. This FIA dataset can be downloaded from http://www.models.kvl.dk/Flow_Injection.(Accessed on 1 October 2021) In FIA, a chemical sample is introduced into the carrier fluid. As it moves downstream, the sample disperses into the carrier reagent and can undergo chemical changes. These changes can be monitored continuously by UV spectroscopy. FIA data is then reflecting the changes over time of the light absorbance of the sample constituents at each of the measured wavelengths. The samples analyzed have mixtures of 2-, 3-, and 4-hydroxy-benzaldehydes (HBA) at different concentrations. All three analytes have different UV absorption spectra, and due to the pH gradient applied during the FIA diffusion, the three chemical compounds are distributed as different acid-base (protonated/unprotonated) species and the experimental UV spectra of the twelve samples analyzed by FIA changed, reflecting these pH changes. Thus, the three-way dataset **X** has in this case 89 rows (FIA flow times), 100 columns (UV wavelengths) and 12 data slices, one for each analyzed sample. Due to the pH gradient, the dataset has a complex spectral shift pattern which is attempted to be modelled using different three-way data analysis strategies. This dataset has been already analyzed using different chemometric methods based on the trilinear model such as PARAFAC, PARALIND (Parallel profiles with linear dependencies) [62,63], and DNTD [32].

## 4. Results

In the different MCR models tested in this work, the criterion for the initial selection of the number of components was the size of singular values of the analyzed dataset. Only those components with singular values larger than those associated to experimental noise were considered. For the ALS optimization, initial spectra were obtained using the purest variables approach [44]. MCR-ALS using the trilinearity constrain, with (1,1,1) or without synchronization (2,2,2) reached convergence in a few number of iterations (<50 iterations). In contrast, MCR bilinear solutions usually did converge more slowly due to its tendency to data overfitting and to include correlated noise into the solutions. Non-negative constraints were imposed on all cases for all profiles. The three-way data structure and adequacy of the trilinear model were also investigated by the SVD of the augmented matrices in their three modes, **X_caug_**, **X_raug_** and **X_saug_**.

### 4.1. Synthetic LC-DAD Dataset

In Figure 3A, SVD plots of **X_caug_**, **X_raug_** and **X_saug_** obtained for the synthetic LC-DAD data set are given. Three larger singular values are clearly distinguished in the plots of **X_caug_** (blue line) and **X_saug_** (black line), in agreement with the number of components used to build this data synthetic LC-DAD set. The spectra and sample profiles of the three components are different and linearly independent as shown in Figure 3B,C. Observe that, starting at the forth singular value, the sizes of the next ones are very small and do not change much anymore, i.e., they are at the experimental noise level. In contrast, the SVD plot of **X_raug_** (red line) shows the presence of two larger additional singular values before reaching the noise level. This augmentation of the chemical rank in this data mode is due to the non-synchronized profiles of the three components, as shown in Figure 2A. As said above, this compels the strict fulfillment of the trilinear model and requires its relaxation to take care of the shifting of the profiles in this mode.

A summary of the results of the application of the different methods in the resolution of this synthetic LC-DAD data set is given in Table 1. This comparison includes different parameters such as the lack of fit and the explained variances, lof and R^2^ (Equations (7) and (8)), which evaluate how well the different methods explain the analyzed data set. Additionally, for each of the three components of the data set, the recoveries of the profiles in the three modes, elution spectra, and sample, are estimated using the correlation coefficients (Equation (9)), and angles between them and the theoretical ones (Equation (10)). In the case of the elution profiles of the resolved components, they are compared in the unfolded augmented mode (see Method section).

PARAFAC2 and MCR-ALS trilinear allowing profile shifting (2,2,2) are given the best results, with explained variances around 99.4%, and lack of fit below 8%, and good recoveries of the profiles in the different modes, with correlation coefficients close to one and small angles <5 degrees. The MCR-ALS bilinear fit is also very good, but the resolved elution profiles for components 2 and 3 were not so well recovered because of the presence of rotation ambiguities when only non-negativity constraints are applied. MCR-ALS trilinear (1,1,1) with synchronized profiles and, especially PARAFAC, fit the data more poorly (lack of fits up to 19% and 24% respectively), and produced inadequate recoveries of the augmented elution profiles of any of the three components (with angles > 20). These worse results are due to the wrong modelling of the shifted elution profiles, not conforming with the imposed trilinear model in both cases. Even worse results were obtained by the ATLD method, with very poor data fit (lack of fit of 35.5%) and even worse recoveries of the elution profiles of two of the components. DNTD produced intermediate results, but it needed a lot of computer memory and resources, with much longer execution times of close to 30 min. The application of DNTD suffers from the way of sorting factors and in the extensive computer resources required (more than 16 GB memory for 600 variables) and it did not recover properly the elution and spectra profiles of component 1. In a previous work [32], DNTD, ATLD, PARAFAC, PARAFAC2 and MCR-ALS bilinear were also compared. Similar conclusions were obtained for DNTD, ATLD, PARAFAC and PARAFAC2. However, as we have shown in the work, MCR-ALS (2,2,2,), i.e., using the trilinearity constraint without synchronization, did provide optimal results, similar to those obtained by PARAFAC2, as it was also already shown in a previous work [27].

Figure 4 shows graphically the comparison of the resolved profiles in the three data modes, elution time, spectra and sample, obtained by ATLD (Figure 4A), PARAFAC (Figure 4B), PARAFAC2 (Figure 4C), DNTD (Figure 4D), MCR-ALS trilinear (2,2,2) (Figure 4E), and MCR-ALS bilinear (Figure 4F). Since ATLD and PARAFAC are based on the strict fulfillment of the trilinear model, the recovered profiles in the different modes were wrong, especially in the sample and elution time modes. Spectra and sample profiles resolved by MCR-ALS bilinear were very similar to the correct ones, but elution profiles were not so well resolved, with double peaks and a considerable amount of noise embedded on them. In contrast, profiles resolved by PARAFAC2 and MCR-ALS trilinear (2,2,2) in the three modes were correctly resolved. Previous works based on the peak shifting correction of the raw data have been given in Ref [64]. The strategy explained in this paper is different, and it implies the flexible implementation of the trilinearity constraint by synchronization of the MCR-ALS resolved profiles in the factor matrix C during the application of the trilinearity constraint at each iteration of the ALS optimization.

### 4.2. Wine GC-MS Experimental Dataset

This wine GC-MS data example was already presented in the literature [31,36,65] to illustrate how to handle the problems derived from the lack of synchronization (peak shifting) of the elution profiles in gas chromatography with mass spectrometric detection. [25,26,28,29]. In this example, a small-time window of the wine GC-MS data was analyzed, where two chemical compounds and a background/solvent contribution were coeluting. Two chemical compounds, 3-hydroxy-2-butanone and hexyl acetate were identified from their mass spectra and a third one was assigned to a solvent/background contribution (see below).

In Figure 3B, SVD plots of **X_caug_**, **X_raug_** and **X_saug_** obtained for this wine GC-MS data set are given. Three larger singular values are clearly distinguished in the plots of **X_caug_** (blue line), in agreement with the number of components of the system. The mass spectra of these three components are different and linearly independent and SVD of **X_caug_** gives the right chemical rank for this system. Observe that, starting at the forth singular value, the sizes of the next singular values are small and do not change significantly any more, i.e., they are at the experimental noise level. In contrast, the SVD plot of **X_raug_** (red line) again shows the presence of two or three additional larger (up to six) singular values before reaching the noise level. This augmentation of the chemical rank in this data mode is due again, as in the previous example, to the non-synchronized profiles of the elution profiles of the components of the system, as shown below. On the other side, the SVD plot of **X_saug_** (black line) shows a complex pattern, with second and third singular values with similar large sizes and decreasing then more slowly up to six or more components. The reason for this behavior should be explained in terms of how the total concentration profiles of the three components change over the different analyzed samples. The best augmented mode to analyze this data system is the spectral mode (blue line), which reflects the chemical nature of the system.

The results obtained with the different variants of MCR-ALS compared with those obtained by ATLD, PARAFAC, PARAFAC2 and DNTD are given in Table 2. ATLD, PARAFAC, PARAFAC2 and DNTD were applied using their default settings, and MCR-ALS was applied using different model configurations. In Table 2, the results obtained by these methods are given. Recoveries of the mass spectra profiles of the two identified compounds, 3-hydroxy-2-butanone, and hexyl acetate, could be evaluated using their reference mass spectra downloaded from Massbank [60], (http://www.massbank.jp/Index, accessed on 1 October 2021). Results for the recovery of the unknown third component associated to the background/solvent contribution are not given, but they are discussed below in Figure 5 and Figure 6.

Data fitting results were slightly worse for ATLD, PARAFAC and MCR-ALS trilinear model (1,1,1) and (1,1,0), due to the large peak sifts between samples in the elution profiles of the two sample constituents and to the difficulty in modelling them with a trilinear model using three components (R^2^ of 94.0%, 94.9%, 95.0% and 95.0%, and lof of 24.6%, 22.6%, 22.2% and 22.8%, respectively). PARAFAC2, DNTD, and MCR-ALS mixed trilinear with elution profiles not synchronized, models (2,2,0) and (2,2,1) and MCR-ALS bilinear (0,0,0) were giving good data fitting results (R^2^ above 99% and lof < 10%). Resolution of the mass spectra of the two sample constituents, 3-hydroxy-2-butanone and hexyl acetate, was reasonably good by all the tested methods, either assuming or not the full trilinear model. The lack of synchronization of the elution profiles did not affect the correct resolution of the mass spectra profiles even if the applied trilinear model was not fulfilled by the data. Recovery of the profiles in the other two data modes (elution and sample profiles) are not given in Table 2 because in this case there are no reference elution profiles to compare (they are unknown in this experimental data set). However, what can be compared instead are the multiple sample augmented elution profiles incorporating the peak shifts in the different chromatographic runs, when they are resolved by the different methods (see Appendix A). For instance, in the case when the unfolded elution profiles obtained by PARAFAC2 and MCR-ALS mixed model (2,2,1) were compared, the correlation coefficients (r^2^) and angles between the augmented elution profiles were very good for the first, secondnd and third (background) components, which were 0.9884 and 8.8, 0.9914 and 7.5, and 0.9996 and 1.7, respectively. Also, when the same comparison is performed for DNTD and MCR-ALS trilinear (2,2,1), the correlation coefficients (r^2^) and angles were good: 0.9857 and 9.7; 0.9829 and 10.6; and 0.9996 and 1.7, as well as for the comparison between PARAFAC2 and DNTD elution profiles, which were 0.9948 and 5.81, 0.9908 and 7.76, 1.0000 and 0.46. This means that three methods were given very similar results and explained the data similarly. If the same comparisons are done with the other trilinear model based methods (PARAFAC, ATLD, and other MCR-ALS (1,1,1) and (1,1,0) models, the results were not so good, because there was more disagreement between the compared profiles, with angles above 20 in most of the cases (see Appendix A). This simply reflects that the elution profiles of the components resolved by these methods based on the strict fulfillment of the trilinear model cannot tackle the lack of synchronization of the elution profiles of the same component in the different chromatographic runs due indeed to the strong time shifting of their peaks.

In Figure 5, the elution (left), spectra (middle), and sample (right) profiles of the MCR-ALS mixed model (2,2,1) are plotted. The elution profiles of 3-hydroxy-2-butanone and hexyl acetate (red and blue) were modelled, allowing for peak shifting during the application of the trilinearity constraint, but the third background component (black) was modelled to strictly fulfill the trilinear condition (the same shape and synchronization, without shift correction). The obtained result (see lower Figure 5 left in black) is reasonable because the background signal is present along the whole chromatographic run and its evolution for the different samples only differs in its total intensity among the different runs in an unknown amount. The mass spectra of the two chemical compounds are well resolved (plots in the middle) and are in agreement with the reference spectra (see also in Table 2, r^2^ and angle values as commented above). Interestingly, again, the mass spectrum resolved for the third background/solvent component (bottom in the middle in black) has well-resolved signals which are clearly differentiated from the mass spectra of the other two chemical compounds. In fact, the mass units (*m/z*) of the major signals of the mass spectrum obtained by the MCR-ALS model (2,2,1) for this background component were 4 (He_2_), 18 (H_2_O), 28 (N_2_) and 32 (O_2_). These signals agree well with the gas composition of the mobile phase. This result was not achieved so well with the other tested models, where the MS signals of the background could not be detangled from the MS signals of the two other chemical compounds and with its elution profile not clearly showing the patterns shown in Figure 5.

In Figure 6, the elution profiles of the three components in the 44 different samples obtained by PARAFAC2, DNTD and the other variants of MCR-ALS are plotted overlaid with the same time axes. These elution profiles can be compared with the elution profiles resolved by the MCR-ALS (2,2,1) model previously shown in Figure 5 (left hand plots). Elution profiles in the different runs of the two chemical compounds (3-hydroxy-2-butanone, and hexyl acetate) were well resolved, allowing for their large peak shits, except when they were modelled assuming the trilinear model with the MCR-ALS (1,1,1). In the case of PARAFAC2 and DTND, the elution profiles were normalized, and they do not reflect the changes of their concentrations in the different samples (they are in the third mode sample profiles), whereas in the different versions of MCR-ALS, they have been plotted without normalization and do reflect these sample concentration changes. Especially interesting is the investigation of the shapes of the background elution profiles in the different chromatographic runs (third component, in the right of Figure 6) resolved using the different MCR-ALS mixed models. In the case of MCR-ALS bilinear modelling of the third component background profile in the different runs, models (0,0,0), (1,1,0), and (2,2,0), the peak shapes feature of the resolved profiles were opposite to the peak shapes resolved for the two profiles of the other two chemical compounds in the different samples. Also, large offsets were observed among the different chromatographic runs. Due to the flexibility of the bilinear modelling and to the rotation ambiguities associated to this type of model [66], the profile of this background component was resolved as a linear combination of the two other component elution profiles and of the constant background profile, which then reflected the shape of the former. As a conclusion of all the tested models, the MCR-ALS mixed model (2,2,1) is therefore the one giving the more reliable results because it is an optimal fit with the data (Table 2), it models the shifts of the profiles of the two chemical compounds (keeping still their same shape) correctly, and it models the background signal offsets of the third component in a very plausible way. The application of trilinearity to this component avoided the presence of rotation ambiguities and was better at resolving the natural features of this background gas mobile phase contribution. When trilinearity was not applied to this component (as in model 2,2,0), its resolution was affected by some linear combination with the other component profiles and mixed with some embedded (correlated) noise, as is shown in Figure 6 (left in black for models (0,0,0) and (2,2,0)). In Appendix A the complete set of elution, spectra and sample profiles obtained in the analysis of the wine GC-MS dataset by the different methods (apart from MCR-ALS mixed model (2,2,1)) already given in Figure 5 are shown for comparison.

### 4.3. Flow Injection Analysis (FIA) Experimental Data

The FIA data set is an example of experimental data with strong linear dependences among the profiles of the components. In fact, these linear dependencies come from the acid-based chemical speciation derived from the pH gradient used during FIA. Due to these pH changes, the three chemical constituents (2HBA, 3HBA and 4HBA, see Section 3.3) were distributed in more than one acid-base species, with different UV spectra. This is a challenge for the trilinear models because the profiles of the acid-base species depend on the pH values and the shapes of these profiles will depend on the synchronization of the pH gradient among them. Curiously, in the literature ([32,61,62]) this system has been considered to be a three-component system, since three were the chemical constituents of the analyzed mixtures, but in fact there could be up to six chemical species (the protonated and unprotonated forms of the three chemical constituents) with different UV spectra because of the acid-base equilibria and pH changes during the FIA process. This fact challenges the application of the trilinear model decomposition using only three components, and other non-trilinear decomposition methods were considered instead [12,13,27,28,30,31]. To illustrate this situation, in this work, the FIA dataset was investigated using the different MCR-ALS bilinear and trilinear model variants, with and without time synchronization of the FIA diffusion (first mode) profiles, and the results were also compared with those obtained by ATLD, PARAFAC, PARAFAC2, and DNTD using a different number of components.

When this data system was analyzed by SVD (Figure 3C), the number of singular values larger than the noise level was clearly higher than three in the spectral mode, (**X_caug_**), giving six larger singular values (the seven was already at the experimental noise level), in agreement with the six acid-base species corresponding to the unprotonated and protonated forms of the three acid compounds. In contrast, the SVD of **X_raug_** and **X_saug_** showed only three larger components (the forth singular value was already very low and at the noise level). This would reflect that in this case, the FIA (time) and sample profiles of the two acid-base species of the same chemical component are linearly dependent and produced the annihilation of the chemical rank of this system in these two data modes. According to these results, it was clear that the mixture system could not be properly explained with only three components, and that a larger number of components are needed. In fact, the SVD of the individual acid-based systems (results not shown) confirmed that the number of species for each of the chemical compounds (2HBA, 3HBA and 4HBA) was two, and that when these three compounds are mixed, the number of components needed to explain the spectral changes should increase to six.

Since the total concentrations of the three chemicals (2HBA, 3HBA and 4HBA) in some of the samples (calibration samples, [32,61,62] were known, they can be used as reference profiles in this sample mode for the comparison of the results obtained by the different tested methods. Moreover, assuming that there was no chemical interaction between the three acid-base constituents, the spectra and FIA profiles of the sample constituents could also be estimated from the individual analysis of the samples containing only one of the three chemical constituents (2HBA, 3HBA or 4HBA). The spectra and FIA profiles obtained in the individual analysis of each chemical constituent were then compared with those obtained in the analysis of their ternary mixtures.

In Table 3, the comparison of the results obtained by different methods considering six (Table 3) species are given for the analysis of the 12 mixture samples. PARAFAC and PARAFAC2 fitted the original data rather well (R^2^ > 99%, see Table 3), but in some cases they were not given good recoveries (r^2^ < 0.9 and angles > 20) for some of the profiles of the six acid-base species profiles. Worse were the results for ATLD and DNTD, with total wrong recoveries. In contrast, the three variants of MCR-ALS (bilinear (0,0,0), trilinear (1,1,1) and trilinear with shift correction (2,2,2)) fitted the data very well, but more importantly, they recovered both the sample and spectra profiles for the six acid-base species well. Notice that in this case (see below Figure 7), the sample profiles of the two acid-base species of the same component would have practically the same sample profile since the species distribution with pH is independent of the total concentration. This is the reason why the MCR-ALS trilinear model (1,1,1) was also giving rather good results in this case. The situation is totally different when only three components are considered.

In Figure 7, FIA, spectra and sample profiles of the six components obtained by using the MCR bilinear model with only non-negativity constraints are shown. This bilinear model gives a good fitting of the original data and explains the FIA process well. The FIA, spectra, and sample profiles of the two species of 2HBA (in red), of 3HBA (in blue), and of 4HBA (in black) were well resolved separately. The spectra profiles (on the middle of the plots) match very well with the two known acid-base spectra of the compounds (broken lines colored in magenta). On the other hand, the sample profiles (on the right of the plots) also match the reference sample profiles well (also broken lines colored in magenta), reflecting, moreover, that for each of the three chemical compounds, the two acid-base species have practically the same sample profile, which is understandable according to pH acid-base equilibria properties and dependencies of these two species from their total concentrations. FIA profiles (on the left of the plots), show only small shifts and shape changes occurring during the analysis of the different samples, which explains why the trilinear model variants of MCR-ALS and PARAFAC and PARAFAC2 worked also quite well for the FIA data, with six components according to the results given in Table 3. In Appendix A, the FIA, spectra and sample profiles obtained by the different methods (apart from MCR-ALS bilinear) are also given. As it is shown graphically in these Figures, in most of the cases, the disagreement between the profiles in the three modes obtained by these methods and the reference ones are larger than for the ones obtained by the MCR-ALS bilinear model given in Figure 6.

When the different methods were compared using only three components (see Appendix A and Appendix A), DNTD and PARAFAC2 were giving good data fits and recoveries of the sample profiles of the three chemical constituents (2HBA, 3HBA and 4HBA) in the different analyzed samples. However, the spectra profiles recovered by all methods using only three components were wrong and not reasonable (see Appendix A). The reason for this is that the models with only three components could not tackle the complexity of the system, particularly in the spectral mode, and therefore their spectra profiles could not be resolved properly.

## 5. Conclusions

Implementation of the trilinearity constraint in MCR-ALS can be performed in a flexible way, tackling different three-way data scenarios where the fulfillment of the trilinear model is challenging. One of the most common situations where departures from the trilinear model occur is when the chromatographic elution profiles of the chemical constituents analyzed in the different samples (or chromatographic runs) are time-shifted. In this work, the results obtained in the analysis of different three-way datasets confirmed that the proposed flexible implementation of the trilinearity constraint provided better results than the stringent application of the trilinear model. The implementation of the shift correction constraint during the ALS iterations is a good alternative to the direct preprocessing correction of these shifts in the raw data. Apart from the lack of synchronization (peak shifting) in the resolved profiles of the components, other problems can be encountered when the shapes of these profiles also change or when there are linear dependencies among the profiles of different components. In these more complex scenarios, even though the data are three-way, the application of the trilinear model is too demanding and the resolution and recovery of the profiles is not realistic. In these cases, the analysis of the three-way data set can still be performed satisfactorily using the bilinear model factor decomposition via matrix augmented data sets along the data mode (usually the spectral one), where the same vector space is shared among the different data slices. Although the uniqueness of the factor decomposition associated to the application of the trilinearity model may be lost under these circumstances, the application of the MCR-ALS bilinear is still possible, giving realistic and meaningful results, especially if other constraints like local rank, selectivity or unimodality can also be applied. The LC-DAD, GC-MS and FIA datasets investigated in this work proved the efficiency and flexibility of the trilinearity constraint implementation in MCR-ALS in these different data scenarios.

## Figures and Tables

**Figure 1 molecules-27-02338-f001:**
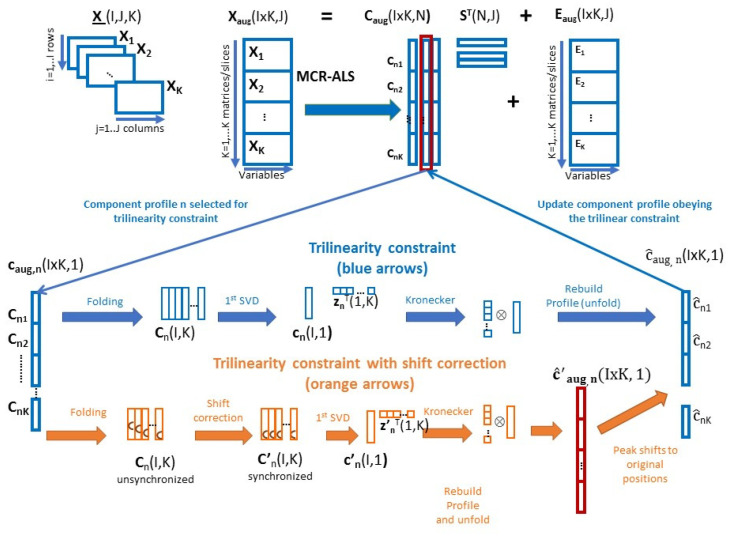
Implementation of the trilinearity constraint with and without synchronization in the MCR-ALS algorithm (see also Equations (4)–(6) in Section 2.2 and Appendix B).

**Figure 2 molecules-27-02338-f002:**
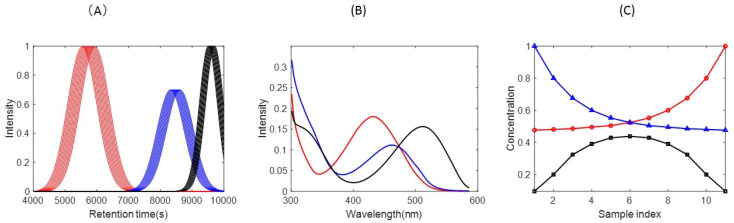
(**A**) Elution (left), (**B**) spectral (middle), and (**C**) sample (right) profiles of three-way dataset 1. Observe the large shifts in the peaks of the elution profiles of the three components in the different chromatographic runs (middle plots). The profiles of the three components are shown in different colors.

**Figure 3 molecules-27-02338-f003:**
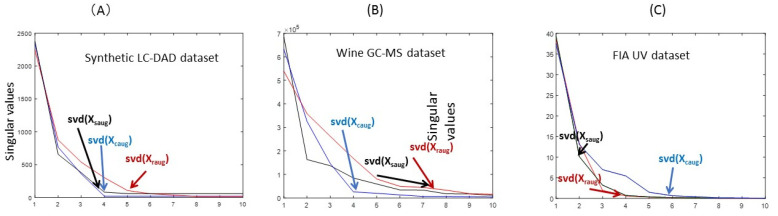
SVD results in the analysis of the three augmented data matrices **X_caug_**, **X_raug_** and **X_saug_** along the three data modes for the three investigated data examples: (**A**) Synthetic LC-DAD dataset; (**B**) Wine GC-MS data set; (**C**) FIA UV dataset.

**Figure 4 molecules-27-02338-f004:**
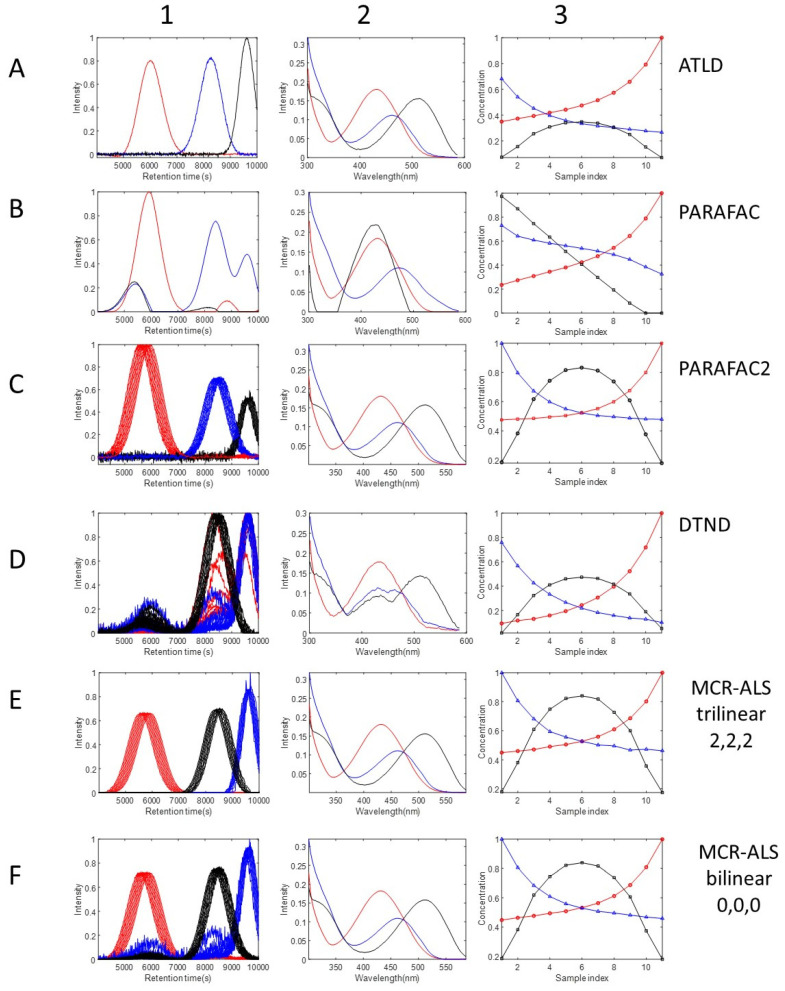
Elution, spectra, and sample profiles resolved by ATLD (**A1**–**A3**), PARAFAC (**B1**–**B3**), PARAFAC2 (**C1**–**C3**), DNTD (**D1**–**D3**), MCR-ALS mixed trilinear (2,2,2) (**E1**–**E3**), and MCR-ALS bilinear (0,0,0) (**F1**–**F3**) in the analysis of the synthetic LC-DAD dataset. See Section 4.1 and Table 1.

**Figure 5 molecules-27-02338-f005:**
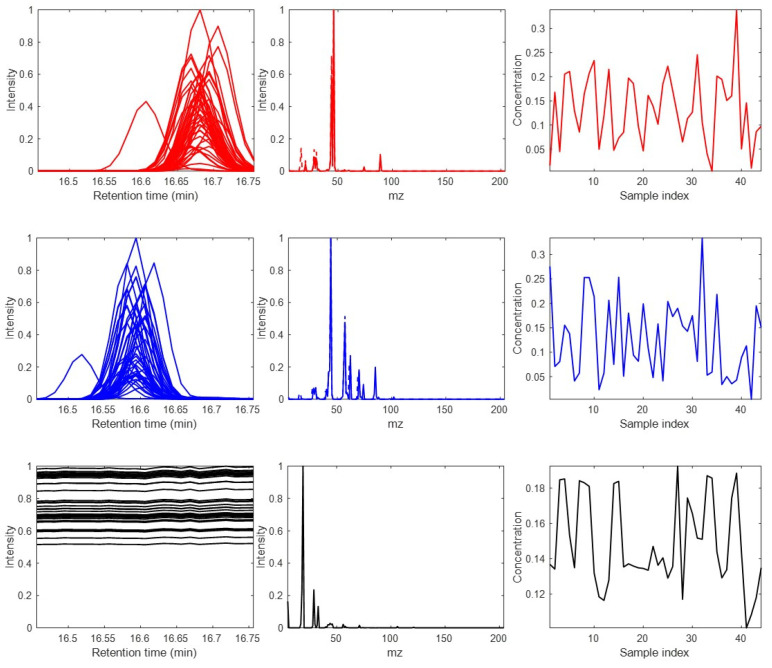
Elution, spectra, and sample profiles of the three components resolved by MCR-ALS mixed trilinear model 2,2,1 in the analysis of the wine GC-MS dataset.

**Figure 6 molecules-27-02338-f006:**
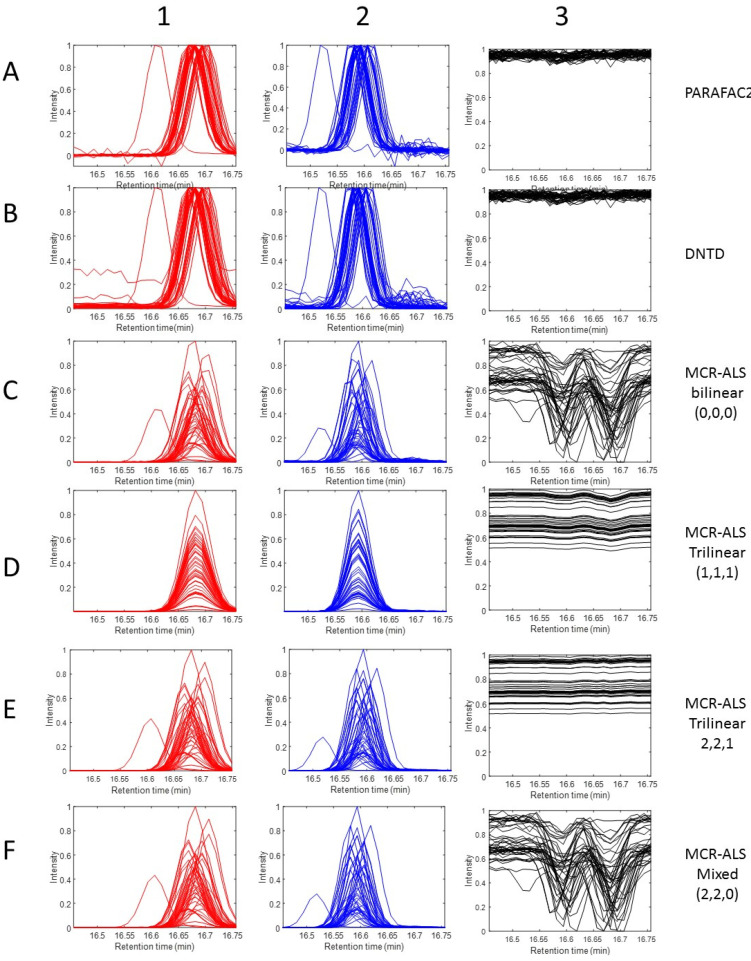
Elution profiles of the three components (3-hydroxy-2-butanone, hexyl acetate and solvent background) in the analysis of the wine GC-MS dataset resolved by PARAFAC2 (**A1**–**A3**), DNTD (**B1**–**B3**), and different MCR-ALS variants: MCR-ALS bilinear (0,0,0) (**C1**–**C3**), MCR-ALS trilinear (1,1,1) (**D1**–**D3**), MCR-ALS trilinear (2,2,2) (**E1**–**E3**), and MCR-ALS mixed (2,2,0) (**F1**–**F3**). See results in Section 4.2 and Table 2.

**Figure 7 molecules-27-02338-f007:**
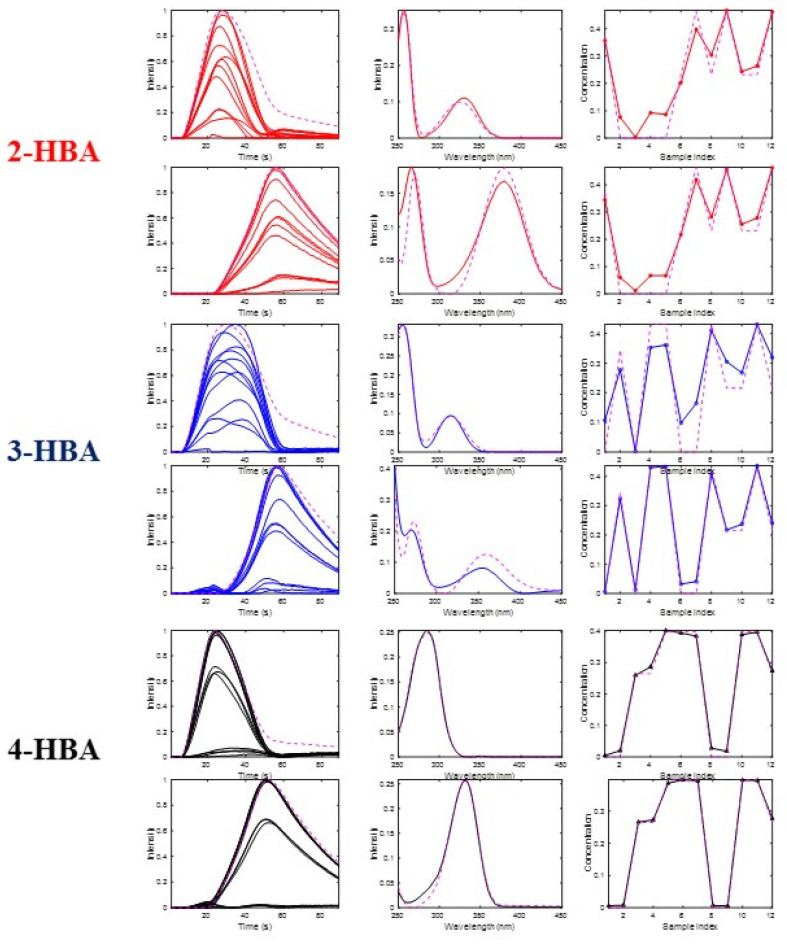
FIA (left), spectra (middle), and sample (right) profiles of the six components resolved in the analysis of the pH gradient FIA-UV dataset by MCR-ALS bilinear, corresponding to the two acid-base species of 2-HBA (red), 3-HBA (blue) and 4-HBA (black). Species spectra and sample profiles were compared with reference ones (cyan). See results in Section 4.3 and Table 3.

**Table 1 molecules-27-02338-t001:** Comparison of results obtained by ATLD, PARAFAC, PARAFAC2, DNTD and different variants of MCR-ALS in the analysis of the LC-DAD data set. In bold red are worse recovered profiles with r^2^ values below 0.9. See Equations (7)–(10) for the meaning of R^2^, lof, r^2^ and angle. See end of Section 2.2 for the meaning of the different MCR-ALS variants.

Methods	R^2^	Lof	Profiles	Recovery of Profiles					
				Component 1		Component 2		Component 3	
				r^2^	Angle	r^2^	Angle	r^2^	Angle
ATLD	87.4	35.5	Elution	** 0.6369 **	** 50.4 **	** 0.07 **	** 85.4 **	0.959	16.4
			Sample	1.0000	0.17	1.0000	0.23	1.0000	0.4
			Spectra	0.9961	5.0	0.9984	3.2	0.9999	0.6
PARAFAC	94.1	24.2	Elution	** 0.7673 **	** 39.8 **	** 0.5486 **	** 56.7 **	0.9128	24.1
			Sample	0.9834	10.4	0.9908	7.8	0.9035	25.4
			Spectra	0.9991	2.4	0.9885	8.7	0.9053	25.1
PARAFAC2	99.4	7. 7	Elution	0.9997	1.3	0.9991	2.4	0.9970	4.5
			Sample	1.0000	0.3	1.0000	0.18	1.0000	0.03
			Spectra	0.9999	0.8	1.0000	0.26	1.0000	0.2
DNTD	99.4	7.5	Elution	** 0.1174 **	** 83.3 **	0.9908	7.8	0.9781	12.02
			Sample	0.9992	2.3	0.9869	9.3	0.9710	13.8
			Spectra	0.9186	23.3	0.9450	19.1	0.9950	5.7
MCR-ALS bilinear (0,0,0)	99.4	7.7	Elution	0.9992	2.3	0.9988	15.0	0.9216	22.8
			Sample	0.9999	0.7	0.9998	1.2	1.0000	0.4
			Spectra	0.9998	1.3	0.9995	1.8	0.9998	1.1
MCR-ALS trilinear (1,1,1)	96.4	18.8	Elution	0.9769	12.3	0.9845	10.1	0.9889	8.5
			Sample	0.9999	0.8	0.9998	1.2	1.0000	0.5
			Spectra	1.0000	0.2	1.0000	0.4	0.9999	0.7
MCR-ALS trilinear (2,2,2)	99.4	7.7	Elution	0.9990	2.5	0.9970	4.4	0.9876	9.0
			Sample	0.9999	0.7	0.9998	1.1	0.9999	1.0
			Spectra	0.9994	1.9	0.9995	1.8	0.9993	2.1

**Table 2 molecules-27-02338-t002:** Comparison of results obtained by ATLD, PARAFAC, PARAFAC2, DNTD and different variants of MCR-ALS in the analysis of the wine GC-MS dataset. In bold red are worse recovered profiles with r^2^ values below 0.9. See Equations (7)–(10) for the meaning of R^2^, lof, r^2^ and angle. See end of Section 2.2 for the meaning of the different MCR-ALS variants.

	R^2^	Lof	Recovery of Spectra Profiles			
			3-Hydroxy-2-Butanone		Hexyl Acetate	
			r^2^	Angle	r^2^	Angle
ATLD	94.0	24.6	0.9855	9.8	0.9673	14.7
PARAFAC	94.9	22.6	0.9869	9.3	0.9793	11.7
PARAFAC2	99.7	5.4	0.9868	9.3	0.9790	11.8
DNTD	99.7	5.4	0.9857	9.7	0.9536	17.5
MCR bilinear (0,0,0)	99.7	5.3	0.9860	9.6	0.9487	18.4
MCR trilinear (1,1,1)	94.8	22.2	0.9866	9.4	0.9544	17.4
MCR mixed (1 1 0)	94.8	22.8	0.9867	9.4	0.9492	18.3
MCR mixed (2 2 0)	99.2	9.4	0.9864	9.5	0.9489	18.4
MCR mixed (2 2 1)	99.2	9.2	0.9865	9.4	0.9790	11.8

In all MCR models, the order of the components is 3-hydroxy-2-butanone, hexyl acetate, and solvent background).

**Table 3 molecules-27-02338-t003:** Comparison of results obtained by ATLD, PARAFAC, PARAFAC2, and different variants of MCR-ALS using six components in the analysis of the FIA dataset. In bold red, the worse recovered profiles with r^2^ values below 0.9 are shown. See Equations (7)–(10) for the meaning of R^2^, lof, r^2^ and angle. See end of Section 2.2 for the meaning of the different MCR-ALS variants.

	R^2^	Lof			Recovery of Sample Profiles					
					FIA Profiles		Spectra Profiles		Sample Profiles	
					r^2^	Angle	r^2^	Angle	r^2^	Angle
ATLD	67.6	56.9	2HBA	acid	** 0.7607 **	** 40.5 **	** 0.8081 **	** 36.1 **	** 0.8081 **	** 36.1 **
				alkali	** 0.9188 **	** 23.2 **	** 0.6934 **	** 46.1 **	** 0.6934 **	** 46.1 **
			3HBA	acid	** 0.8738 **	** 29.1 **	** 0.8254 **	** 34.4 **	** 0.8254 **	** 34.3 **
				alkali	** 0.2152 **	** 77.6 **	** −0.8030 **	** 143.4 **	** −0.8030 **	** 143.4 **
			4HBA	acid	** 0.8874 **	** 27.4 **	** 0.6834 **	** 46.9 **	** 0.6834 **	** 46.9 **
				alkali	** 0.7951 **	** 37.3 **	** −0.3078 **	** 107.9 **	** −0.3078 **	** 107.9 **
DNTD	38.1	65.5	2HBA	acid	** 0.6910 **	** 46.3 **	** 0.0912 **	** 84.8 **	0.9676	14.6
				alkali	0.9206	23.0	** 0.4533 **	** 63.0 **	0.9623	15.8
			3HBA	acid	0.9208	23.0	** 0.8561 **	** 31.1 **	** 0.8125 **	** 35.7 **
				alkali	** 0.8293 **	** 34.0 **	** 0.4030 **	** 66.2 **	** 0.7944 **	** 37.4 **
			4HBA	acid	** 0.7811 **	** 38.6 **	** 0.1393 **	** 82.0 **	** 0.6932 **	** 46.1 **
				alkali	0.9366	20.5	** 0.8108 **	** 35.8 **	0.9840	10.3
PARAFAC	99.1	9.4	2HBA	acid	0.9772	12.3	0.9940	6.3	0.9770	12.3
				alkali	0.9990	2.5	0.9852	9.9	0.9473	18.7
			3HBA	acid	0.9727	13.4	0.9708	13.9	0.9354	20.7
				alkali	0.9807	11.3	** 0.4535 **	** 63.0 **	** 0.6755 **	** 47.5 **
			4HBA	acid	0.9980	3.7	0.9798	11.5	0.9855	9.8
				alkali	0.9942	6.2	0.9776	12.1	0.9647	15.3
PARAFAC2	99.9	0.8	2HBA	acid	0.9417	19.7	** 0.4128 **	** 65.6 **	0.8836	27.9
				alkali	0.8311	** 33.8 **	0.9548	17.3	0.9705	14.0
			3HBA	acid	0.9661	15.0	0.9857	9.7	0.9063	25.0
				alkali	** −0.4305 **	** 115.5 **	0.9115	24.3	0.9482	18.5
			4HBA	acid	** 0.8031 **	** 36.6 **	** 0.8508 **	** 31.7 **	0.9537	17.5
				alkali	** −0.0725 **	** 94.2 **	0.9163	23.6	0.9914	7.5
MCR-ALS bilinear (0,0,0)	99.9	1.2	2HBA	acid	0.9956	5.4	0.9747	12.9	0.9895	8.3
				alkali	0.9703	14.0	0.9994	2.0	0.9572	16.8
			3HBA	acid	0.9980	3.6	0.9807	11.3	0.9976	4.0
				alkali	0.9339	21.0	0.9974	4.1	0.9988	2.8
			4HBA	acid	0.9997	1.3	0.9949	5.8	0.9997	1.3
				alkali	0.9970	4.4	0.9992	2.4	0.9997	1.3
MCR-ALS trilinear (1,1,1)	99.4	7.4	2HBA	acid	0.9681	14.5	0.9967	4.7	0.9806	11.3
				alkali	0.9997	1.3	0.9696	14.1	0.9918	7.3
			3HBA	acid	0.9694	14.2	0.9973	4.2	0.9519	17.0
				alkali	0.9928	6.9	0.9244	22.4	0.9962	4.9
			4HBA	acid	0.9897	8.2	0.9981	3.6	0.9996	1.5
				alkali	0.9992	2.3	0.9976	3.9	0.9998	1.1
MCR-ALS trilinear (2,2,2)	99.9	3.1	2HBA	acid	0.9784	11.9	0.9995	1.7	** 0.8881 **	** 27.3 **
				alkali	0.9991	2.4	0.9688	14.3	0.9981	3.5
			3HBA	acid	0.9751	12.8	0.9892	8.4	0.9761	12.5
				alkali	0.9957	5.3	0.9815	11.1	0.9947	5.9
			4HBA	acid	0.9909	7.7	0.9002	25.8	0.9932	6.7
				alkali	0.9932	6.7	0.9972	4.3	0.9998	1.2

## Data Availability

LC-DAD synthetic data set is available under request to one of the authors, XZ. The other two data sets are available in web pages http://www.massbank.jp/Index (wine data set) and http://www.models.kvl.dk/Flow_Injection (flow injection data set). Accessed on 1 October 2021.

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
