# Peer review of "Flexible Implementation of the Trilinearity Constraint in Multivariate Curve Resolution Alternating Least Squares (MCR-ALS) of Chromatographic and Other Type of Data"

_molecules, 2022, doi:10.3390/molecules27072338_

Round 1

Reviewer 1 Report

Please find the reviewers's comments in attachment. 

Author Response

See in the attached file

Reviewer 2 Report

In the manuscript “Flexible implementation of the trilinearity constraint in Multivariate Curve Resolution Alternating Least Squares (MCR-ALS) of chromatographic and other type of data” presents an interesting variant of Multivariate Curve Resolution Alternating Least Squares for data analysis (LC-DAD among others) comparing with other algorithms (ATLD, PARAFAC, PARAFAC2 and DNTD).

In this work, the proposed implementation of the trilinearity constraint in MCR-ALS was studied in three different three-way data scenarios. The results obtained confirmed that the proposed flexible implementation of the trilinearity constraint provided better results than the rigorous trilinear model. Many advantages have been observed such as: (I) Implementation of displacement correction constraint during ALS iterations (II) Peak displacement resolution as well as other problems can be encountered when the shapes of these profiles also change or when there are dependencies linear between the profiles of different components. It was also proven that the application of bilinear MCR-ALS allows obtaining significant results, especially if other restrictions such as local rank, selectivity or unimodality are additionally applied.

In view of the above and the results found by the authors, I am in favor of the publication.

Minor revisions:

  • In line 216 complete: singular value decomposition (SVD).
  • Lines 464 to 468 correct: The elution profiles of these three compounds at the 600 measured retention times in each sample are plotted in Figure 2a, showing the presence of very large shifts in their peak maxima. The UV spectra of the three components are plotted in Figure 2b, in the wavelength range between 300 and 580 nm with one nm resolution. Eleven samples with three different compounds were analyzed 468 by LC-DAD.
  • In (3.2. Wine GC-MS experimental data) check the page address because I couldn't access the link provided on line 472.
  • In lines 518-519, what is described in the text does not correspond to the information presented in the Figures. Please correct Figures 3b and 3c, or the text.
  • In line 626 change “22,6.2%” to “22.6%”.
  • I suggest enlarging the graphs in Figure 7.

Author Response

See in the attached file

Round 2

Reviewer 1 Report

Please find the reviewer's comments in attachment. 
